# Isolation and Characterization of a Lytic *Vibrio parahaemolyticus* Phage vB_VpaP_GHSM17 from Sewage Samples

**DOI:** 10.3390/v14081601

**Published:** 2022-07-22

**Authors:** Xunru Liang, Yuhang Wang, Bin Hong, Yanmei Li, Yi Ma, Jufang Wang

**Affiliations:** 1School of Biology and Biological Engineering, South China University of Technology, Guangzhou 510006, China; bixrliang@mail.scut.edu.cn (X.L.); biwangyuhang@mail.scut.edu.cn (Y.W.); 202010108485@mail.scut.edu.cn (B.H.); 202010108442@mail.scut.edu.cn (Y.L.); bimayikobe@scut.edu.cn (Y.M.); 2Guangdong Provincial Key Laboratory of Fermentation and Enzyme Engineering, South China University of Technology, Guangzhou 510006, China

**Keywords:** *Vibrio parahaemolyticus*, phage, vB_VpaP_GHSM17, characterization, genome analysis

## Abstract

*Vibrio parahaemolyticus* is a major foodborne pathogen and the main cause of diarrheal diseases transmitted by seafood such as fish, shrimp, and shellfish. In the current study, a novel lytic phage infecting *V. parahaemolyticus*, vB_VpaP_GHSM17, was isolated from the sewage of a seafood market, Huangsha, Guangzhou, and its morphology, biochemistry, and taxonomy features were identified. Morphological observation revealed that GHSM17 had an icosahedral head with a short, non-contractile tail. The double-stranded DNA genome of GHSM17 consisted of 43,228 bp with a GC content of 49.42%. In total, 45 putative ORFs were identified in the GHSM17 genome. Taxonomic analysis indicated GHSM17 belonging to genus *Maculvirus*, family *Autographiviridae.* In addition, GHSM17 was stable over a wide range of temperatures (20–60 °C) and pH (5–11) and was completely inactivated after 70 min of ultraviolet irradiation. The bacterial inhibition assay revealed that GHSM17 could inhibit the growth of *V. parahaemolyticus* within 8 h. The results support that phage GHSM17 may be a potential candidate in the biological control of *V. parahaemolyticus* contamination in aquaculture.

## 1. Introduction

*Vibrio parahaemolyticus* is a Gram-negative halophilic bacterium that is widely found in oceans and estuaries. Eating any raw seafood containing pathogenic *V. parahaemolyticus* can cause human infection, which is characterized by watery diarrhea, abdominal cramps, nausea, vomiting, and headache [1]. Most clinically-derived pathogenic *V. parahaemolyticus* can produce two major toxins, thermostable direct hemolysin (TDH) [2] and TDH-related hemolysin (TRH) [3]. Both toxins have the characteristics of hemolytic toxicity, enterotoxigenicity, cardiac toxicity, and cytotoxicity that can affect the infected hosts [4,5,6]. Additionally, *V. parahaemolyticus* can also lead to severe acute hepatopancreatic necrosis disease (AHPND) in shrimp aquaculture, which causes huge economic loss to shrimp farmers [7]. At present, for the rapid and effective control of the proliferation of *V. parahaemolyticus*, antibiotics are widely and inevitably used in aquafarming, the transportation and sales section, leading to multiple antibiotic-resistant strains in pathogenic communities [8]. Microbial risk assessments indicated that both ready-to-eat (RTE) foods and aquatic products in China were at great risks of infection with pathogenic *V. parahaemolyticus* [9,10]. In addition, most isolated *V. parahaemolyticus* strains showed a certain degree of resistance to streptomycin, cefazolin, and ampicillin [11], posing particularly serious threats and challenges to human public health and economic problems worldwide [12,13,14]. Notably, biological additives, rather than antibiotics, will be a potential candidate for the prevention and control of the pathogenic bacteria in aquatic products [15].

Bacteriophages are universal viruses on earth that can specifically recognize, capture, and lyse their host bacteria in various environments. With the inevitable increase of multiple antibiotic-resistant bacteria and the lack of novel antibiotics, phage therapy is becoming increasingly popular due to its enormous potential to destroy bacteria [16,17]. As early as 1919, phage therapy was successfully applied to chickens infected with *Salmonella Gallinarum* [18]. At present, phages are widely used as a biological control agent in the food and agriculture industries, wastewater treatment, and aquaculture [19]. Previous studies have shown that the direct application of a single phage or phage cocktail to RTE foods and meat could significantly reduce contamination with various food-borne pathogens [20]. Lytic phages were applied to chicken skin to reduce the number of *Salmonella Enteritidis* or *Campylobacter jejuni* by 2 log CFU within 48 h [21]. A study was the first to report that a cocktail of three phages (ECP-100) significantly reduced *E. coli* O157:H7 on lettuce and cantaloupe [22]. Phage therapy was also conducted in vivo, demonstrating that phages could control bacterial infections in fish, shrimp, and other aquatic products in aquaculture [23,24]. In hatchery trials, the phage treatment of *Penaeus monodon* larvae infected *Vibrio harveyi* resulted in a larval survival rate of over 85%, suggesting that phages would be an effective alternative to antibiotics [25]. Importantly, these viruses were nontoxic to humans and could eradicate bacterial biofilms due to biofilm-degrading enzymes, which had also proven to be more promising than antibiotics [26].

In this study, a lytic phage infecting *V. parahaemolyticus*, termed as vB_VpaP_GHSM17, was isolated from the sewage of a seafood market, Huangsha, Guangzhou, and its morphology, biochemistry, and taxonomy features were identified. Phage GHSM17 was a new member of the genus *Maculvirus*, family *Autographiviridae*, and its growth inhibitory effect on *V. parahaemolyticus* was investigated. The characterization and analysis will potentially be of great help in further expanding our cognition of *Vibrio* phages and also provide a new tool for *V. parahaemolyticus* control.

## 2. Materials and Methods

### 2.1. Bacterial Strains and Phage Isolation

A total of 17 strains of *V. parahaemolyticus* from Guangzhou Customs Technology Center were used in this study, which were isolated from import and export seafood. The genotypes and sources of each strain are shown in Table 1. All *V. parahaemolyticus* were stored in 25% (*v*/*v*) glycerol at −80 °C and further validated by PCR to detect species–specific genes (*tlh*) and virulence genes (*tdh* and *trh*), according to a previous study [27,28].

The sewage sample was collected from Huangsha, Guangzhou, and processed as previously described for the phage culture [29]. After the sample was filtered, MgSO_4_ was added and completely dissolved. The mixture was filtered through a filter membrane again. The filter membrane was eluted for 10 min. Then, the eluted liquid was mixed with 100 μL of *V. parahaemolyticus* OY49 into 5 mL of Tryptic Soy Broth (TSB, 2 mM CaCl_2_) and cultivated at 37 °C, 220 rpm for 24 h. After culturing for 24 h, the mixed culture was centrifuged at 8000× *g* for 10 min at 4 °C, and the supernatant was filtered through a 0.22-μm filter to remove residual bacteria. The phage proliferation was repeated three times, and the presence of phages was verified by a double-layer plate. Briefly, the phage at the appropriate dilution (100 μL) was mixed with *V. parahaemolyticus* OY49 (100 μL) in 5 mL of TSB (0.4% agar), incubated at 37 °C, and plaques were observed after 3 h. Three candidate plaques were picked by sterile pipette tips into TSB (2 mM CaCl_2_). Then, 100 μL of *V. parahaemolyticus* OY49 was added, and the mixture was cultured at 37 °C with 220 rpm for 24 h. The cultured mixture was centrifuged, and the presence of plaques was observed by a double-layer plate. The phage purification step was repeated six times by a double-layer plate. Finally, the purified phages were stored in 30% (*w*/*v*) glycerol at −80 °C.

### 2.2. Phage Morphological Observation

A fresh phage suspension was prepared, and polyethylene glycol 8000 (PEG 8000) was added to the phage suspension with a final concentration of 15%. The mixture was incubated at 4 °C overnight. After incubation for 24 h, the deposit was centrifuged at 12,000× *g* for 20 min at 4 °C and then resuspended with SM buffer (1 mL) to prepare the phage suspension concentrate for electron microscopy analysis. Three gradients (1.3, 1.5, and 1.7 g/mL) of CsCl solutions were prepared. The phage suspension was subjected to CsCl gradient centrifugation at 170,000× *g*, 4 °C for 2 h, and the blue band containing phages was collected immediately. Phage concentrates were placed on carbon membranes prepared with 3% (*w*/*v*) phosphotungstic acid. Morphological features were observed under a transmission electron microscope (TEM, Talos L120C) at 80 kV.

### 2.3. Extraction and Sequencing of Phage Genome

After the phage concentrate was obtained according to Section 2.2, the DNA was extracted from phage GHSM17. DNase I and RNase A were used to digest bacterial DNA, and a phage DNA extraction kit (Leagene Biotechnology, Beijing, China) was used for purification. DNA quantification, electrophoresis, enzymatic digestion, and other general manipulations were performed according to standard procedures [30]. DNA concentration was measured by Nano Vue Plus spectrophotometer (GE Healthcare, Chicago, IL, USA). Purified DNA was sent to Personal Biotechnology Corp. (Shanghai, China) for whole-genome sequencing. A DNA library was prepared with the insert size of 400 bp for the phage sample. Phage genome samples were sequenced by the Illumina MiSeq platform using the PE 2 × 150 bp strategy.

### 2.4. Genome Assembly, Annotation, and Comparison

The raw sequencing data contained some low-quality reads with adapters. In order to ensure the quality of the subsequent information analysis, *SOAPec* [31], was used to filter all reads based on the *Kmer* frequency. The filtered high-quality reads were reassembled via *A5-MiSeq* v20160825 [32] and *SPAdes* v3.12.0 [33]. The sequences were extracted, and the sequences with high sequencing depth were compared with the National Center for Biotechnology Information Non-redundant (NR) Database by *blastn* [34]. The viral genome sequences of each splicing result were picked out. *Pilon* v1.18 [35] was used to correct the results for obtaining the final genome sequence. After obtaining the whole genome sequence, *Diamond* v0.8.36 [36] and *GeneMarkS* v4.32 [37] were used for the prediction of open reading frames (ORFs) in the phage genome. Finally, a circular map of the genome of phage GHSM17 was generated via *CGView*. In addition, predictions of non-coding RNAs were mainly obtained by comparison with the *Rfam* v14.1 database [38].

After obtaining the assembled genome sequence, it was uploaded to the NCBI database for the sequence alignment. In brief, the most similar species sequences and taxonomic names were obtained through sequence alignment. Firstly, the *blastn* [34] was used to align the target sequences in the database. The calculation function of the alignment algorithm used default parameters. After obtaining the most similar known sequence information, the results were filtered through the three thresholds of *Query Coverage* (90~100%), *Percent Identity* (50~100%), and *E value* (0~1 × 10^−6^) to obtain the final alignment result. Secondly, after obtaining the preliminary taxonomic results, the information obtained from the comparison was retrieved in the ICTV to obtain all known species (that is, taxonomically classified species) and genome information of the genus. The genome information of all known species from this genus was downloaded from the database, and then the ORF-based genome-wide structure and function linear alignment was performed. The results were visualized using *Easyfig* [39]. Thirdly, after finishing the preliminary genome-wide alignment of the target sequence, the initial taxonomic information of the phage strain was obtained, including the three levels of family, genus, and species. All the relevant phage genome information of this family was downloaded from the NCBI database for *fastANI* analysis [40]. Based on the analysis results, we obtained the average nucleotide identities of the phage GHSM17 and all related species to further determine its taxonomic status, with parameters: *cutoff_down =* 0.75, *fragLen =* 3000, *clustering_distance_cols/rows = euclidean*. *Pheatmap* [41] was used to visualize the calculation and analysis results. Finally, based on the predicted ORF structure and gene function, we selected the sequences corresponding to the two molecules of the RNA polymerase and terminase large subunit for phylogenetic tree analysis. *MEGAX* was used to analyze the phylogenetic tree with the following parameters: *No. of Bootstrap Replications =* 1000, *Substitution Model/Method = Kimura2-parameter model*, *Gaps/Missing Data Treatment = Use all sites*, *Branch Swap Filter = Very Strong* [42].

### 2.5. Phage Titer Determination

Phage titers were determined by the double-layer plating. Briefly, after 100 μL of phage suspension appropriately diluted and 100 μL of *V. parahaemolyticus* OY49 in mid-log phase were poured into double-layer plates, the mixture was incubated at 37 °C for the observation and counting of plaques after 3 h. The appropriate dilutions were determined by three replicate plates for each dilution. Finally, the phage titer was calculated as follows: plaque number × 10 × dilution gradient = pfu/mL.

### 2.6. Determination of Phage Host Range

To determine the host range of the purified phage, a spot assay was used for 17 *V. parahaemolyticus* strains (Table 1). Using the double-layer plating, 100 μL of *V. parahaemolyticus* in mid-log phase and 6 mL of TSB (0.4% agar) were mixed and poured onto double-layer plates. After solidifying for 5 min, 2 μL of phage suspension in TSB was added dropwise to the plate. The plate was then incubated at 37 °C. After 3 h, it was checked whether a zone of inhibition occurred. It was judged based on the clarity of the zone of inhibition: clear lysis zone (+) and no lysis zone (−).

### 2.7. Determination of Multiplicity of Infection (MOI) and One-Step Growth

To determine the optimal MOI, 100 μL of bacteriophage and 100 μL of *V. parahaemolyticus* OY49 (1 × 10^8^ cfu/mL) in 10 mL of TSB (2 mM CaCl_2_) were prepared according to the ratio of MOIs of 0.0001, 0.001, 0.01, 0.1, 1, 10, and 100, respectively. The mixture was incubated at 220 rpm for 10 h at 37 °C and centrifuged at 8000× *g* for 5 min at 4 °C. The supernatant was taken and filtered through a 0.22-μm filter to obtain a phage suspension. Phage titers were determined by double-layer plating. The multiplicity of infection with the highest titer was the best MOI.

The one-step growth curve was determined according to the method of Yang et al. [43]. The *V. parahaemolyticus* OY49 suspension cultured to OD_600_ = 1 was diluted to 1.0 × 10^8^ cfu/mL. An amount of 1 mL of bacteria was taken and centrifuged at 8000× *g* for 5 min at room temperature, the pellet was resuspended in 0.9 mL of SM buffer (100 mM NaCl, 8 mM MgSO_4_, and 50 mM Tris-HCl, 0.01% Gelatin, pH 7.5), and phage suspension was added according to the ratio of MOI of 0.1 for mixing. The mixture was incubated for 10 min at 37 °C. After centrifugation at 8000× *g* for 5 min at room temperature, the pellet was suspended in 10 mL of TSB. The mixture was then incubated with shaking at 37 °C and 220 rpm. Phage titers were determined by double-layer plating every 10 min.

### 2.8. Stability of Phage to Environmental Stress

Tests for phage stability under particular environmental conditions, including temperature, pH, and UV, followed a previously-described protocol [44]. An amount of 1 mL of phage suspension (4 × 10^7^ pfu/mL) was added to 1.5-mL centrifuge tubes, which were respectively incubated at different temperature ranges from 20 °C to 90 °C for 1 h. For pH stability experiments, the pH of TSB was adjusted to 3, 4, 5, 6, 7, 8, 9, 10, and 11, respectively. An amount of 1 mL of phage suspension (1 × 10^9^ pfu/mL) was added into 9 mL of TSB with a specific pH value and incubated in a water bath at 37 °C for 1 h. The UV tolerance experiment was carried out as follows: the phage suspension (1 × 10^8^ pfu/mL) was added to the sterile plate and irradiated by UV at 30 W/100 W power in a biosafety cabinet. The sample was about 15 cm away from the UV lamp, and a 100-μL sample was drawn every 10 min. The double-layer plate method was used to determine the phage titer.

### 2.9. Phage Control Experiment

In order to detect the potential of application, the antibacterial effect of phage GHSM17 against *V. parahaemolyticus* was studied in a 96-well plate with *V. parahaemolyticus* OY49 and SC123 according to a previous study [45]. Briefly, 100 μL of bacterial dilution (1 × 10^8^ cfu/mL) was added to 5 mL of TSB, and phage suspension was added into the experimental groups as MOI 0.01, 0.1, 1, and 10. An amount of 200 μL of the mixture was pipetted into a 96-well plate. The blank group was without neither *V. parahaemolyticus* nor phage suspension, and the control group was with *V. parahaemolyticus* and without phage suspension. The 96-well plate was placed in the Microplate Reader (37 °C, 220 rpm) for culture. Starting from 0 h, the absorbance value (OD_600_) was detected for each sample every 1 h.

### 2.10. Statistical Analysis

According to the principles of standard control experiments, the experiment was repeated three times, and the results were expressed in the form of mean ± standard deviation (*SD*). *GraphPad Prism* 8.0.1 software was used to analyze the above statistics and results.

## 3. Results

### 3.1. Morphology and Host Range of Bacteriophage GHSM17

In this study, the phage GHSM17 was isolated from sewage using the double-layer plating. Clear plaques (about 2 mm in diameter) were formed on a double-agar plate (Figure 1A,B), indicating that the phage was lytic, and it also was demonstrated by mitomycin C treatment (Appendix A). The TEM revealed that the phage GHSM17 consisted of an icosahedral head and a short noncontracted tail (Figure 1B). Head length, head diameter, and tail length were measured at 54 ± 2, 53 ± 2, and 19.5 ± 1 nm (n = 2) by TEM, respectively. Referring to the official guidelines of the International Committee on Taxonomy of Viruses (ICTV) and the International Virus Classification and Nomenclature, the phage GHSM17 belonged to the order *Caudovirales* and then was named as vB_VpaP_GHSM17.

Seventeen strains of *V. parahaemolyticus* were used to determine the host range of the phages. As the results, phage GHSM17 was able to lyse 9 of the 17 tested strains, with a relatively broad host range (Table 1), suggesting the potential of the GHSM17 to be a candidate for phage therapy. Different phages with the same host were isolated, with vB_VpS_BA3 having a quite narrow host spectrum, infecting only 5 (5/61) strains of *V. parahaemolyticus*, while vB_VpS_CA8 was able to lyse 22 (22/61) strains [43]. The host range of different phages was determined by the total number of hosts it used. Therefore, the host spectrum of phages from different sources cannot be directly compared. However, phages with a broader host spectrum mean that they may be the better candidates for phage therapy.

### 3.2. Optimal MOI and One-Step Growth Curve

As shown in Figure 2A, the phage GHSM17 infected *V. parahaemolyticus* OY49 at the MOI of 0.0001, 0.001, 0.01, 0.1, 1, 10, and 100 and showed high bacteriophage titer, reaching the maximum phage titer at the MOI of 0.1. The life cycle of bacteriophage, including the latent period, explosive phase, and plateau phase, was quantified using a one-step growth curve. As seen from Figure 2B, GHSM17 was characterized by a relatively short latency period of 20 min, a rise phase of 100 min, and a burst size of 316 pfu/cell, indicating that phages grew efficiently and rapidly after adsorption on the host surface. Taking these together, phage GHSM17 showed excellent lytic effect on the *V. parahaemolyticus* isolate.

### 3.3. Stability of Bacteriophages

The primary conditions for further applications of phages were investigated by resistance to the environmental stress assay. Heat stability tests showed that phage GHSM17 was stable below 60 °C but completely inactivated at 80 °C (Figure 3A). Furthermore, as shown in Figure 3B, phage GHSM17 maintained high activity from pH 5.0 to pH 11.0, and the activity dramatically decreased when the pH was lower than 5.0, while it was essentially inactive at pH 3.0. The titer of GHSM17 declined every 10 min under UV irradiation (Figure 3C), achieving extermination after 80 min of exposure to UV at 30 W. By increasing the intensity of the UV light, the death of the phage was accelerated, and the time was shortened to 70 min. The results demonstrated that the phage could withstand various stresses in an aquaculture environment.

### 3.4. Bacterial Inhibition Assay

The growth-inhibitory effect of phage GHSM17 on host bacteria was assessed using the bacterial inhibition assay by *V. parahaemolyticus* OY49 and *V. parahaemolyticus* SC123. *V. parahaemolyticus* was cultured in TSB (3% NaCl) at 37 °C and infected at MOIs of 0.01, 0.1, 1, and 10. It can be seen in Figure 4 that *V. parahaemolyticus* OY49 and SC123 reached a stable phase after 14 h of culture, respectively. After the addition of the phage, *V. parahaemolyticus* OY49 was significantly inhibited in growth within 8 h, while an exponential phase was observed after 8 h (Figure 4A). Meanwhile, after the phage infection, *V. parahaemolyticus* SC123 regrew after 4 h, but its growth was inhibited within 20 h at all MOI ratios (Figure 4B). The number of viable cells was measured at 20 h, and it was found that *V. parahaemolyticus* OY49 regrew rapidly, while *V. parahaemolyticus* SC123 regrew relatively slowly. In conclusion, the growth of *V. parahaemolyticus* OY49 and *V. parahaemolyticus* SC123 were inhibited by phage GHSM17 within 8 and 4 h, respectively; then, *V. parahaemolyticus* OY49 entered the exponential growth phase, while *V. parahaemolyticus* SC123 was still suppressive.

### 3.5. Characteristics of Phage Genome

The Illumina NovaSeq sequencing platform was used to obtain the genomic sequence of GHSM17. After standard sampling and preprocessing procedures, a total of approximately 31,206,584 high-quality reads were obtained with an average length of approximately 400 bp. After automated assembly and manual optimization to further align the entire genome with a single contig, the whole genome of GHSM17 was obtained. The whole genome of phage GHSM17 consisted of linear double-stranded DNA, with a full length of 43,228 bp and a GC content of 49.42% (Table 2).

Genome ends were identified as direct terminal repeat (DTR) sequences by assessing sequence coverage using *PhageTerm*. The entire genome structure of GHSM17 is shown in Figure 5A. A total of 45 ORFs were predicted, 23 of which were similar to genes encoding known functional proteins. All predicted ORFs were divided into six modules (Figure 5B), including DNA metabolism modules (ORF 7, ORF 22, ORF 25, ORF 27, ORF 35, ORF 36, ORF 37, and ORF 38), a lytic module (ORF 8), a packaging module (ORF 6), structural modules (ORF 9, ORF 10, ORF 12, ORF 14, ORF 15, ORF 17, ORF 18, and ORF 19), additional function modules (ORF 4, ORF 21, ORF 23, and ORF 30), and the rest were assigned to hypothetical protein modules (Appendix A). No tRNA genes were found, implying that the replication of phage GHSM17 was highly dependent on the host’s translation machinery.

To determine the overall similarity of phage GHSM17 to the genomes of 51 *Vibrio* phages in *Genbank* (as of March 2022), we calculated the average nucleotide identity (ANI) values. All 51 *Vibrio* phages belonged to the family *Autographiviridae*, and they belonged to the following seven genuses/subfamilies: *Beijerinckvirinae, Colwellvirinae, Cyclitvirus, Maculvirus, Melnykvirinae, Studiervirinae*, and *Tawavirus*. Their genome sizes ranged from ~36 to 47 kb (Appendix A). The clustering results showed that the clustering algorithm based on *euclidean* distance divided 51 phages into 7 clusters, corresponding to *Beijerinckvirinae*, *Colwellvirinae*, *Cyclitvirus*, *Maculvirus*, *Melnykvirinae*, *Studiervirinae*, and *Tawavirus*, and the classification results of the algorithm were completely consistent with the known classification results of the ICTV database (Figure 6). From this reasonable classification result, GHSM17 was verified to belong to genus *Maculvirus* taxonomically. From the heatmap, GHSM17 had the highest homology of 94.82% with vB_VpaP_KF2 (Accession number: NC_048036.1) isolated from Korea. Meanwhile, the results of the genome identity between phage GHSM17 and all phages of the genus *Maculvirus* are shown in Appendix A, which showed that all genome identities (percent identity, P.I.) were less than 95%. All the above results indicate that the phage GHSM17 was a new species belonging to the genus *Maculvirus*.

Species origin and evolution were described by phylogenetic tree. We found that the phylogenetic analysis method based on the terminase large subunit can well resolve the precise location and evolutionary pathway of GHSM17 in the developmental map (Figure 7A). As shown in the phylogenetic tree, GHSM17 was the closest relative to seven *Vibrio* phages, OWB, vB_VpP_FE11, vB_VpaP_KF2, vB_VpaP_KF1, vB_VpP_DE10, vB_VpP_NS8, and vB_VpaP_MGD1. These eight *Vibrio* phages clustered on the same clade (bootstrap value: 96), indicating that GHSM17 belonged to the *Maculvirus* genus of the *Autographiviridae* family. Simultaneously, as another most conserved protein in the phage genome, the developmental tree constructed by the sequence corresponding to RNA polymerase (Figure 7B) still achieved a very significant result (bootstrap value: 90). It was worth noting that, in both trees, the *Pantoea* phage LIMElight, which did not belong to the genus *Maculvirus*, was observed to be very close to the aforementioned clusters, even more similar (bootstrap value: ~72–79) than some phages of the genus *Maculvirus*. This indicated that, in the evolution of related proteins in the *Pantoea* phage, there was a potential homology between the genus *Limelightvirus* and *Maculvirus*.

## 4. Discussion

At present, *V. parahaemolyticus* is the principal pathogen causing food-borne infections in most countries and coastal areas [4]. Food poisoning caused by *V. parahaemolyticus* accounts for 20% to 30% in Japan, and it was the main pathogenic factor of gastroenteritis caused by seafood consumption in Asia and the United States [46,47,48,49]. Phage therapy was considered a promising strategy due to its safety, non-pollution, and low side-effects on aquaculture and humans [50]. Phage products were approved by the FDA as direct food additives and are commercially available [51]. For example, a cocktail of six phages (ListShield) was approved by the FDA, which was effective against 170 *Listeria monocytogenes* strains in RTE foods and poultry products [52]. In this study, the lytic phage vB_VpaP_GHSM17 was isolated from Huangsha sewage in Guangzhou. GHSM17 had an excellent capability to control and kill bacteria efficiently, with rapid recognition and adsorbence to *V. parahaemolyticus* strains.

Phage GHSM17 is a short-tailed phage with an icosahedral head and formed transparent plaques on lawns (Figure 1). *Vibrio* phage vB_VpP_DE17 [53], with head length (47 ± 2 nm) and tail length (17 ± 2 nm), was similar in morphology to the phage in this study. Both of them belonged to *Caudovirales*. In addition, other forms of *Vibrio* phages were isolated from marine, clam, and shrimp, such as long-tailed phages φ-1~4 belonging to *Myoviridae* [54], vB_VpaP_MGD2 belonging to *Podoviridae* [55], and filamentous phage V5 belonging to *Inoviridae* [56], illustrating the morphological diversity of *Vibrio* phages.

Phage GHSM17 is a lytic phage with a relatively broad host range. The host range of each phage was determined by the total number of hosts it uses. It was reported that phage VP06 lysed 16 (28.1%) of 57 *Vibrio* strains, while phage PG07 lysed 47% (14/30) of the tested strains [57]. Therefore, it cannot be directly compared to the host range of different phages. GHSM17 exhibited a 20 min incubation period and a 100 min rising phase, with a lysate size of 316 pfu/infected cell (Figure 2). The one-step growth curve indicated that the phage could grow rapidly and efficiently after being adsorbed to the host surface, reflecting the efficient lysis of phage GHSM17 on *V. parahaemolyticus* isolates. In addition, GHSM17 exhibited broad temperature (20 °C–60 °C) and pH (5–11) stability, similar to other *Vibrio* phages [43], suggesting that the phage retained its activity well at room temperature and a broad pH range.

Bacterial inhibition experiments showed that GHSM17 inhibited *V. parahaemolyticus* within 8 h (Figure 4), making it a potential bio-inhibitor candidate. Although GHSM17 inhibited *V. parahaemolyticus* within 8 h, the inhibiting effect decreased as the bacteria growth resumed after 8 h, which might be due to the selective pressure associated with phage predation leading to the bacterial acquisition of resistance, which was observed in a previous study [58]. Currently, only the lytic phage is available for phage therapies due to its inability to generate horizontal gene transfer between bacteria [59]. Further, to exclude lysogenic and toxic genes, a full sequence analysis is required before the phages are used in the treatment of bacterial diseases.

To further determine the virological classification, the analysis of the genomic information of GHSM17 was performed. Through genome sequencing, the whole genome of GHSM17 was 43.2 kb in length, encoding 45 protein genes (Figure 5A). The 23 ORFs were homologous to genes encoding known functional proteins, including DNA metabolic modules, lytic modules, packaging modules, structural modules, and additional functional modules (Figure 5B). Whole-genome alignment by *blastn* and ANI indicated that GHSM17 was a new member of the *Maculvirus* genus, *Autographiviridae* family. Phages containing RNA polymerases were classified within the *Autographiviridae* family, including *Vibrio* phage GHSM17, vB_VpaS_OWB, vB_VpaP_KF1, vB_VpaP_KF2, and VP93 [60,61,62]. Phylogenetic tree analysis indicated that GHSM17 was closest to the seven *Vibrio* phages OWB, vB_VpP_FE11, vB_VpaP_KF2, vB_VpaP_KF1, vB_VpP_DE10, vB_VpP_NS8, and vB_VpaP_MGD1 (Figure 7). These eight *Vibrio* phages clustered on the same clade, indicating that GHSM17 belonged to the *Maculvirus* genus of the *Autographiviridae* family.

Bacteriophages are extremely specific to the host, with no inhibitory and therapeutic effects on bacteria that cannot be lysed. The host range of bacteriophages was determined by their tail fiber protein [63]. In order to expand the host range of phage therapy, different phages were mixed as a cocktail to reduce the development of phage resistance in bacteria. According to previous studies, the cocktail method reduced the selective pressure exerted by specific phages on their host bacteria [50]. If bacteria resistant to phage attacks emerge, the cocktail can be further modified or improved by adding different phages or replacing existing phages with others [64]. Further studies will be necessary to investigate the safety and efficacy of controlling *V. parahaemolyticus* infection in aquaculture.

## 5. Conclusions

*V. parahaemolyticus* is a pathogenic bacterium commonly found in seawater and seafood. In this work, a novel phage GHSM17 towards this bacterium was isolated and characterized and its potential as an alternative for biological control was also evaluated. GHSM17 had an icosahedral head with a short, non-contractile tail, and it belonged to the family *Autographiviridae*. This lytic phage had the characteristics of a relatively wide host range and high lysis efficiency. Furthermore, the phage was stable over a wide pH and temperature range. Here, we provided the preliminary evidence supporting the control potential of phage GHSM17 against *V. parahaemolyticus*. Future research will focus on characterizing the biosafety of phage in against *V. parahaemolyticus* infection in aquaculture and determining the efficacy of GHSM17 in vivo.

## Figures and Tables

**Figure 1 viruses-14-01601-f001:**
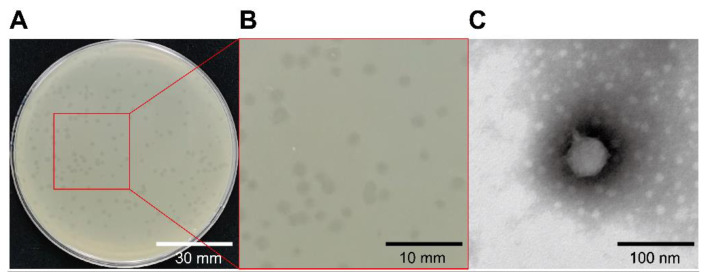
Characterization of vB_VpaP_GHSM17. (**A**,**B**) Phage plaques; (**C**) transmission electron microscopy.

**Figure 2 viruses-14-01601-f002:**
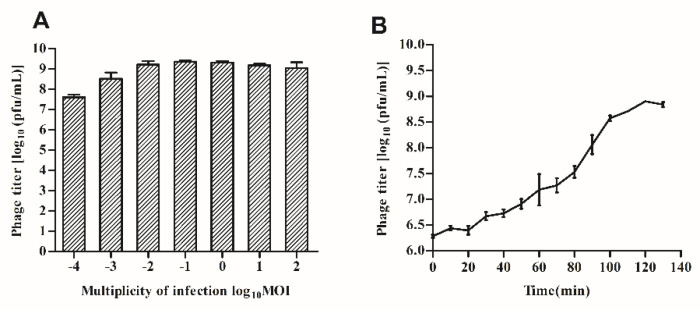
Biological characterization of vB_VpaP_GHSM17. (**A**) MOI; (**B**) one step growth curve.

**Figure 3 viruses-14-01601-f003:**
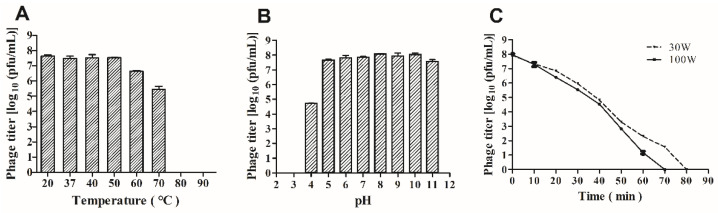
Stability of vB_VpaP_GHSM17 at (**A**) pH; (**B**) temperatures; and (**C**) UV values.

**Figure 4 viruses-14-01601-f004:**
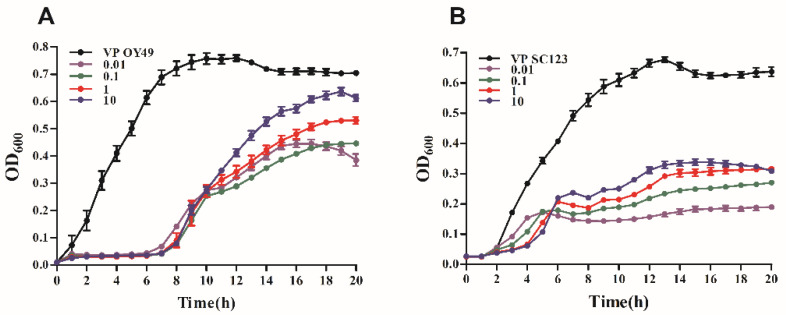
In vitro lysis of vB_VpaP_GHSM17 against (**A**) *V. parahaemolyticus* strain OY49 and (**B**) *V. parahaemolyticus* strain SC123 at multiplicity of infections (MOIs) 0.01, 0.1, 1, and 10 for 20 h.

**Figure 5 viruses-14-01601-f005:**
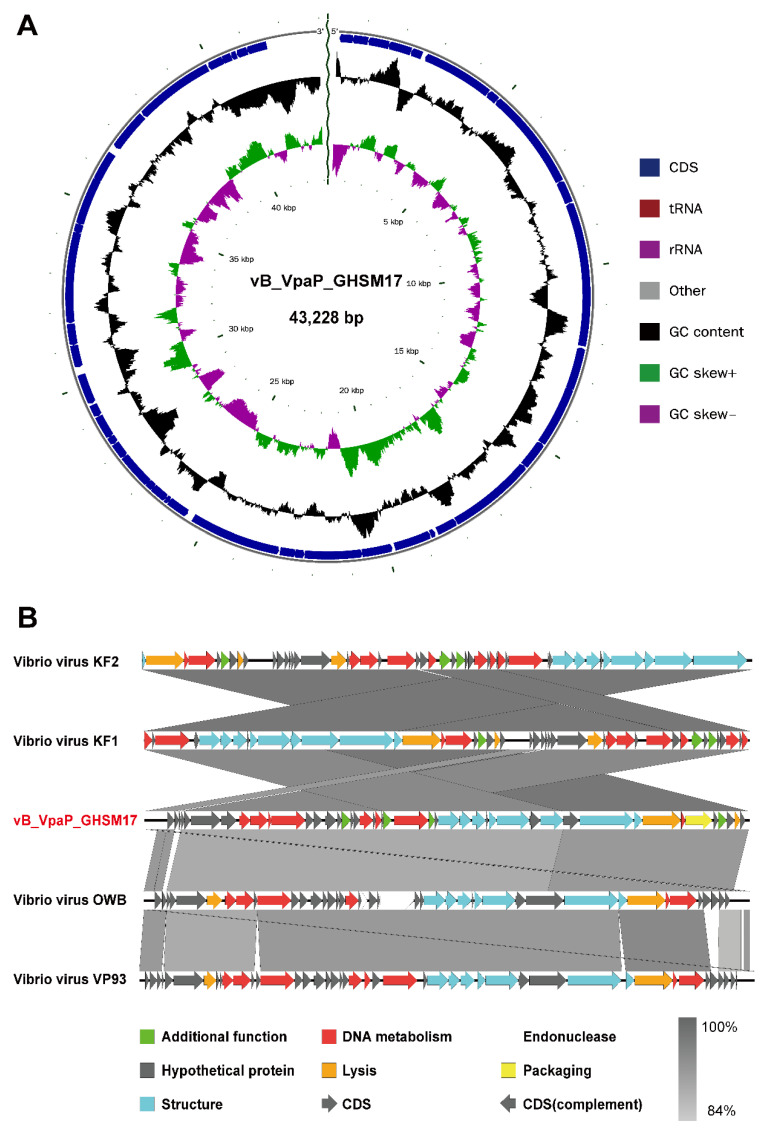
Genomic characterization of vB_VpaP_GHSM17. (**A**) Schematic diagram of GHSM17. From inside to outside, the first to third circles represent the scale, GC Skew, and GC content, respectively; the fourth and fifth circles represent the positions of CDS, tRNA, and rRNA on the genome. (**B**) Schematic representation of the genomic organization of GHSM17 compared to all four classified species in genus *Maculvirus*, family *Autographiviridae*.

**Figure 6 viruses-14-01601-f006:**
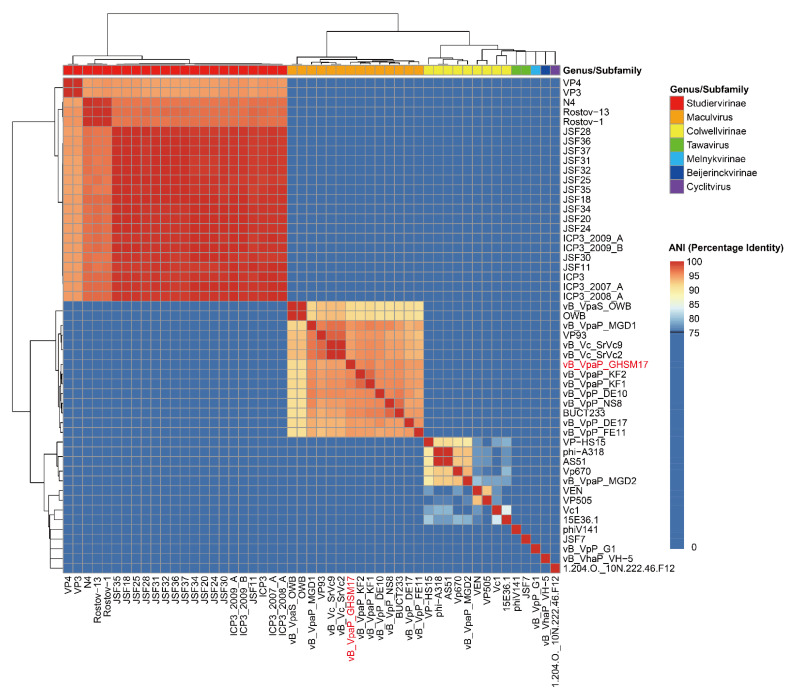
Heatmap of ANIm for 51 *Vibrio* phages. Values range from 0 (0%, navy blue) to 1 (100%, red).

**Figure 7 viruses-14-01601-f007:**
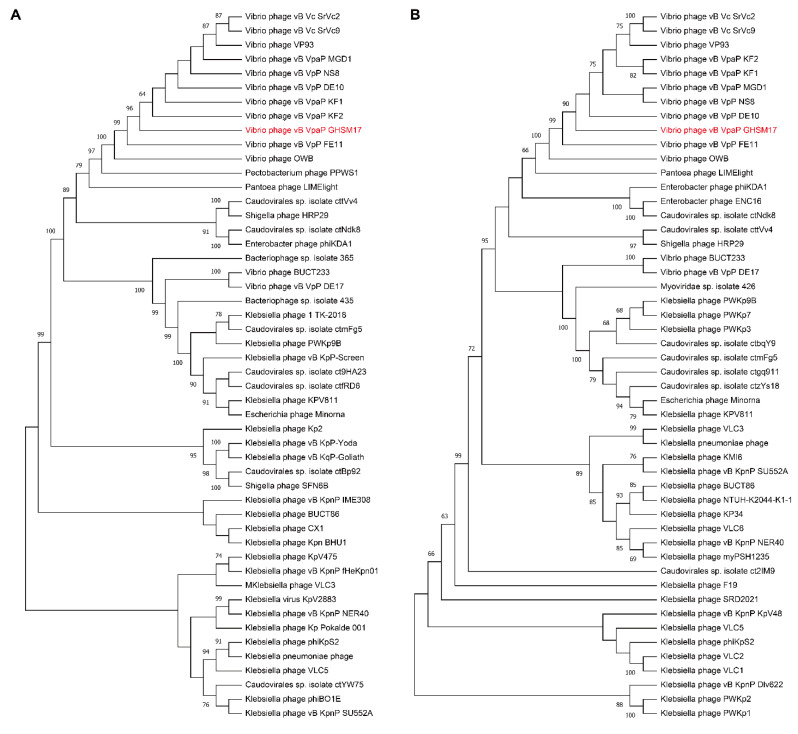
Phylogenetic trees of *Vibrio* phage from the family *Autographiviridae* based on (**A**) the terminase large subunit and (**B**) the RNA polymerase.

**Table 1 viruses-14-01601-t001:** Host range of phage vB_VpaP_GHSM17.

Strain	Origin	Lytic Ability	Virulence Genes
*V. parahaemolyticus* ATCC17802	American Type Culture Collection	+	*Tlh*^+^/*tdh*^−^/*trh*^+^
*V. parahaemolyticus* PR13	Pearl River	−	*tlh*^+^/*tdh*^−^*/trh*^−^
*V. parahaemolyticus* OY14	American Oysters	+	*tlh*^+^/*tdh*^−^*/trh*^−^
*V. parahaemolyticus* FFTF11	Frozen Fork Tail Fillets	+	*tlh*^+^/*tdh*^−^*/trh*^−^
*V. parahaemolyticus* BC21	Bengali Frozen Cuttlefish	−	*tlh*^+^/*tdh*^−^*/trh*^−^
*V. parahaemolyticus* BC20	Bengali Frozen Cuttlefish	+	*tlh*^+^/*tdh*^−^*/trh*^−^
*V. parahaemolyticus* IFH23	Indonesian Frozen Hairtail	−	*tlh*^+^/*tdh*^−^*/trh*^−^
*V. parahaemolyticus* TGS36	Thai Grass Shrimp	+	*tlh*^+^/*tdh*^−^*/trh*^−^
*V. parahaemolyticus* SAL38	South African Lobster	−	*tlh*^+^/*tdh*^−^*/trh*^−^
*V. parahaemolyticus* HA44	Haliotis	−	*tlh*^+^/*tdh*^−^*/trh*^−^
*V. parahaemolyticus* CR48	Crab	+	*tlh*^+^/*tdh*^−^*/trh*^−^
*V. parahaemolyticus* OY49	Oyster	+	*tlh*^+^/*tdh*^−^*/trh*^−^
*V. parahaemolyticus* TMS61	Thai Mantis Shrimp	−	*tlh*^+^/*tdh*^−^*/trh*^−^
*V. parahaemolyticus* SC123	Scallop	+	*tlh*^+^/*tdh*^−^*/trh*^−^
*V. parahaemolyticus* SC126	Scallop	−	*tlh*^+^/*tdh*^−^*/trh*^−^
*V. parahaemolyticus* CR127	Crab	+	*tlh*^+^/*tdh*^−^*/trh*^−^
*V. parahaemolyticus* SE132	Seawater	−	*tlh*^+^/*tdh*^−^*/trh*^−^

**Table 2 viruses-14-01601-t002:** Structural characteristics of phage vB_VpaP_GHSM17 genome.

Phage	Len. (bp)	GC (%)	ORFs	tRNA	Termini	Type
vB_VpaP_GHSM17	43,228	49.42	45	0	direct terminal repeat	linear

## Data Availability

The accession number of sequence data for *Vibrio* phage vB_VpaP_GHSM17 in GenBank is: OM362522. All detailed data presented in this study are available on request from the corresponding authors.

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
