# Peer review of "Isolation and Characterization of a Lytic Vibrio parahaemolyticus Phage vB_VpaP_GHSM17 from Sewage Samples"

_viruses, 2022, doi:10.3390/v14081601_

Round 1

Reviewer 1 Report

The authors present a very good work on the isolation of a vibriophage for an important pathogen, however, in order for it to be accepted and published in Viruses, some minor corrections are required.

Line 166-172: Indicate how many and which strains were used for the host range, if only Vibrio parahaemolyticus was evaluated or if other species were used to determine the range

Line 181-182 The Vibrio parahaemolyticus strain is the same as that used in MOI indicate

Line 184: describe SM buffer formula

Line 189-200: What was the reason why they used different concentrations of vibriophage in each of the tests, were they different stocks?

Line 210: How long was the 1 hour reading taken?

Author Response

Point 1: Line 166-172: Indicate how many and which strains were used for the host range, if only Vibrio parahaemolyticus was evaluated or if other species were used to determine the range.

Response 1: We have indicated all the bacterial species which were used for determining the range in the experimental design section (lines 170-171).

Point 2: Line 181-182: The Vibrio parahaemolyticus strain is the same as that used in MOI indicate.

Response 2: We have added the Vibrio parahaemolyticus strain, which is the same one as in MOI determination (line 186).

Point 3: Line 184: describe SM buffer formula.

Response 3: We have added the SM buffer formula in the experimental design section (lines 188-189).

Point 4: Line 189-200: What was the reason why they used different concentrations of vibrio phage in each of the tests, were they different stocks?

Response 4: The phage suspensions were from the same stock, and they were diluted in different gradients for different tests. For the optimal temperature test, the phage suspension was diluted to 4×107 pfu/mL. In the pH test, 1 mL of phage suspension (109 pfu/mL) was added into 9 mL of TSB to a final concentration of 1×108 pfu/mL.

Point 5: Line 210: How long was the 1 hour reading taken?

Response 5: It took 2 mins for data reading, while the sample was taken every 1 hour.

Reviewer 2 Report

Congratulations to authors for their study, findings, and presentation. I hope in a future study, they will address the apparent resistance of V. parahaemolyticus to this newly discovered phage and find solutions to overcome it.

Author Response

Response: We thank the reviewer for this practical suggestion. We agree that this situation needs to be resolved for further application in the control of V. parahaemolyticus. We will address apparent resistance of V. parahaemolyticus to this newly discovered phage and find solutions to overcome it in future study.

Reviewer 4 Report

 In this study, the authors isolated a novel lytic phage vB_VpaP_GHSM17 infecting Vibrio parahaemolyticus. The authors conducted a series of assays to characterize this novel phage in terms of its morphology, biochemistry and taxonomy features. The stability of this lytic phage was found to be stable over a range of temperature and pH. The authors concluded that this phage could be a potential candidate for the biological control of V. parahaemolyticus.

Overall, the authors did a very comprehensive analysis of this novel lytic phage. This might be out of scope for this study, but have the authors done a cocktail phage inhibition assay on either single or mixed V. parahaemolyticus strains? It will be interesting to study how certain phages might have synergistic effects in inhibiting V. parahaemolyticus. Similarly, if multiple V. parahaemolyticus strains are combined, they can also share cross-protection from each other.

Author Response

Response: We appreciate the reviewer for this practical suggestion. We agree that it would be interesting and valuable to study how certain phages might have synergistic effects in inhibiting V. parahaemolyticus. We will perform a cocktail of phage inhibition assays on single or mixed V. parahaemolyticus strains in subsequent experiments.